# Innate Immune Cells during Machine Perfusion of Liver Grafts—The Janus Face of Hepatic Macrophages

**DOI:** 10.3390/jcm11226669

**Published:** 2022-11-10

**Authors:** Nicola Sariye Roushansarai, Andreas Pascher, Felix Becker

**Affiliations:** Department of General, Visceral and Transplant Surgery, University Hospital Muenster, 48149 Muenster, Germany

**Keywords:** machine perfusion, normothermic, hypothermic, ischemia–reperfusion injury, liver transplantation, clinical trials

## Abstract

Machine perfusion is an emerging technology in the field of liver transplantation. While machine perfusion has now been implemented in clinical routine throughout transplant centers around the world, a debate has arisen regarding its concurrent effect on the complex hepatic immune system during perfusion. Currently, our understanding of the perfusion-elicited processes involving innate immune cells remains incomplete. Hepatic macrophages (Kupffer cells) represent a special subset of hepatic immune cells with a dual pro-inflammatory, as well as a pro-resolving and anti-inflammatory, role in the sequence of ischemia–reperfusion injury. The purpose of this review is to provide an overview of the current data regarding the immunomodulatory role of machine perfusion and to emphasize the importance of macrophages for hepatic ischemia–reperfusion injury.

## 1. Introduction

Liver transplantation (LT) has been established as the only curative therapy for end-stage liver disease [1,2] and the preferred treatment of hepatocellular carcinoma (HCC) [3]. However, in parallel to the undoubted success of LT, an ever-growing dilemma has emerged in the field of transplant medicine: the mounting number of patients awaiting a liver graft that exceeds the number of suitable organs. Thus, organ scarcity has become the cardinal problem in transplant medicine, and various strategies are currently employed to expand the donor pool. One promising tool is machine perfusion (MP), a technique in which liver grafts are continuously perfused (reviewed in in Panconesi et al. (2022) [4]). This innovative technique offers novel perspectives for organ preconditioning as well as viability assessment. Currently, normothermic perfusion (NMP) and hypothermic machine perfusion (HMP) are the most commonly used modalities in clinical practice [5].

While MP has now been clinically established, the medical community is divided regarding its effect on the complex hepatic immune system during perfusion. Moreover, our understanding of the physiologic processes involving innate immune cells is incomplete. In particular, hepatic macrophages (Kupffer cells) have gained attention recently due to their dual role during ischemia–reperfusion injury (IRI). These cells are classic effector cells of the innate immune response and elicit graft damage by participating in the sterile inflammatory response; additionally, they have unique anti-inflammatory and, especially, pro-resolving capacities. This is of special interest in the context of NMP, which introduces a formally unknown phase (ex vivo reperfusion on the device without contact with the recipient’s immune system) to the sequence of IRI. Some key factors such as macrophage polarization and damage-associated molecular pattern (DAMP) [6] circulation also need to be investigated in more detail. Thus, this review provides an overview of innate immune cell interactions in the context of hepatic MP and addresses their specific role as a key target to further optimize MP and, subsequently, improve outcomes for LT.

## 2. MP of Liver Grafts

### 2.1. Basic Concepts and Clinical Implications

The basic principle behind the MP of liver grafts is continuous oxygenated perfusion, which refills the energy supplies of mitochondria and washes out metabolic substrates that accumulate during cold ischemia [7]. This way, the organ’s metabolic demands for oxygen and nutrients are met, and preservation of the microcirculation is granted. MP, therefore, facilitates prolonged organ preservation times, allows viability assessment, and offers a unique opportunity for therapeutic interventions prior to transplantation. MP has shown substantial benefits compared to the conventionally used technique of static cold storage (SCS), especially considering the ever-growing dependence on organs with a higher susceptibility to IRI. These livers from extended criteria donors (ECD) usually combine multiple risk factors such as elevated donor age [8], steatosis (macrosteatosis > 30%, mixed steatosis > 60%), or prolonged cold ischemia time (CIT). In addition to maintaining organ viability, continuous oxygenated perfusion replenishes mitochondrial adenosine triphosphate (ATP) stores [9] and protects the organ by eliminating acidotic waste products. Viability assessment represents another advantage of MP, which allows for detailed organ monitoring and facilitates the choice in favor of or against high-risk liver grafts.

MP can be conducted at various temperatures and ranges from hypothermic (4–10 °C) and sub-normothermic (20–25 °C) to normothermic (35–37 °C) perfusion. Currently, the two commonly used approaches are NMP and HMP.

#### 2.1.1. HMP

Multiple studies have demonstrated the potential of HMP for restoring energy reserves as one of the main mechanisms of action in protecting liver grafts against the detrimental effects of IRI. This is achieved by the induction of hypothermia (4–10 °C), which reduces ATP demand, and additionally oxygenation, which leads to the resuscitation of mitochondria and reduction of reduction of nuclear injury [10,11]. The effects of hypothermia and oxygenated perfusion lead to slowed cell metabolism through the effects of cooling and they rejuvenate ATP resources in mitochondria. Furthermore, immunosuppressive captivities of HMP have been stated to be related to reduced Kupffer cell activity [12]. Both mechanisms ultimately result in the mitigation of IRI and protection of the donor organ. Most common HMP techniques utilize an oxygenated perfusate that is either applied through the portal vein (HOPE) or through both the portal vein and the hepatic artery (D-HOPE) [13]. Compared to SCS, HMP has been demonstrated to be beneficial in reducing ischemic bile duct injury and showed an improved one-year survival rate in livers donated after circulatory death (DCD). In particular, the rate of non-anastomotic biliary strictures (NAS), which represent a major complication of ischemic cholangiopathy after liver transplantation, was significantly reduced after implementation of D-HOPE, as presented in a recent multicenter, prospective, randomized trial [14,15]. Similar data were obtained when prolonged HOPE (>2 h) was compared to DHOPE in a European, multicenter, cohort study [16]. These findings suggest a preventive mechanism of HMP with respect to NAS after LT, which will be discussed further below.

Regarding perfusion time and logistics, HOPE and D-HOPE might be easier to utilize than NMP, but are only at the beginning regarding the assessment of organ viability [17]. Additionally, HMP is used to prolong preservation times in order to facilitate transplantation logistics, since there is the growing evidence that extended HMP application is safe [16,18].

The minimal duration of oxygenated hypothermic perfusion with respect to beneficial effects on mitochondria and ATP resuscitation was previously described as at least 1–2 h [19,20].

After this time, a mitigation of the inflammatory response with a reduction of proinflammatory markers was observed [19]. Interestingly, one hour after HOPE of human liver grafts, a decrease in markers that indicate mitochondrial injury was noted [20]. Additionally, both animal and human liver grafts were investigated after one hour of NMP in this study. In contrast to HOPE, normothermic perfused liver grafts did not show an improvement of mitochondrial injury after one hour. This underlines the common suggestion that NMP requires a longer perfusion time for beneficial effects [21], which is discussed more detailed below. The perfusion time of D-HOPE was safely extended to 24 h, followed by 4 h NMP for viability assessment in a porcine experiment [22]. All liver grafts met viability criteria and did not show an elevation of inflammatory cytokines. Beyond 24 h, further investigations regarding the effect of oxygenated hypothermic liver perfusion are awaited.

#### 2.1.2. NMP

The driving concept behind NMP is providing near physiological conditions by continuous extracorporeal perfusion of the graft at body temperature (37 °C) using an oxygenated carrier solution. Thus, NMP restores cellular metabolism by a multifactored process, including the elimination of citric acid cycle products and improved ATP restoration [23], as well as providing oxygen and nutrition supply [24]. In clinical practice, NMP is also commonly used to determine organ viability, which facilitates the assessment of high-risk organs. Another advantage is that extended preservation times enable LT to be conducted despite logistical barriers.

In clinical practice, erythrocyte concentrates (RBCs, donor matched [23] or blood group O, Rhesus negative 2 [25]) are used as an oxygenated carrier solution for NMP. These contain leukocyte and platelet remnants that undoubtably should be considered in the evaluation of inflammation after MP. RBCs are used since they show better oxygen delivery capacities and less hepatocellular injury compared to other tested organ perfusion solutions (OPS) [26,27,28]. However, OPSs are under current investigation to optimize MP under sub-normothermic temperatures (13–21 °C), with promising results of a synthetic hemoglobin substitute [29]. With respect to liver function and production of pro- and anti-inflammatory cytokines, similar results compared to RBCs were obtained. Yet, a decreased activation of surface markers of resident liver immune cells such as macrophages and dendritic cells could be shown [29]. In addition to that, a modulation of immune response by cell-free oxygen carrier solutions in sub-normothermic MP was observed [30]. Sub-normothermic MP is a variation of perfusion that is performed through the portal vein and hepatic artery [31]. Since little is known about the immunomodulation in the direct comparison of these new OPS and their potential for NMP, further studies regarding the effects of perfusion solutions on donor immune cells are warranted.

Alternative NMP protocols have been described with a controlled oxygenated rewarming (COR) technique, resulting in a D-HOPE–COR–NMP sequence. Recently, studies have reported on the in situ connection of an NMP device in the donor (called ischemia-free organ transplantation (IFLT) [32]), which shows promising results with superior one-year survival rates compared to SCS. Both protocols have not yet been applied to broader use due to limitations, such as logistical and technical complexity. Regarding IRI, NMP has been shown to diminish the progression of inflammation and tissue damage both histologically and molecularly in clinical trials [23,33].

In preclinical porcine studies, the perfusion periods of NMP were set up to 4 [21,34], 5 [35], and 6 [36] h as minimal perfusion times. A reduced inflammatory response was observed after 6–8 h of NMP in porcine animal studies [36,37]. In a small animal model, NMP was reduced to one hour, and liver grafts showed reduced inflammatory markers compared to SCS, followed by transplantation [38], which stays in contrast to recent findings regarding the effects of NMP on mitochondrial injury after one hour perfusion of small animal and human liver grafts [20]. In contrast to HOPE, no optimal perfusion time is suggested for NMP with respect to the inflammatory response. Previously, the evaluation of liver grafts for transplantation that met one or more “high risk” criteria was studied during the minimal perfusion time of 4 h [39]. Within this time, lactate clearance together with at least two more parameters of hepatocellular function indicated whether a liver graft was suitable for transplantation.

Regarding the postoperative survival, a recent study reported an improvement of survival rate after 20 h NMP compared to 5 h NMP in a porcine model [35]. In clinical practice, the median normothermic perfusion time is 8–10 h [23] and does not exceed 24 h due to the risk of hemolysis and to maintain stable glucose metabolism [12]. Within this timespan, a mitigation of IRI was described [33], but there is a need for future investigations regarding a detailed analysis of the temporal dynamics of NMP and the immune response. The maximum perfusion time is challenged by current experimental trials preserving human and porcine livers for 7 days [9] and 12 days [40]. Inflammatory markers and DAMPS showed no increase during 7 days of perfusion in human livers [9]. Liver grafts were perfused with RBCs, while bile production and blood values were used for monitoring. This long-term perfusion was accomplished using an altered perfusion machine with, among others, adjusted perfusion circuit to avoid hemolysis.

However, as highlighted above, NMP offers more advantages than the ability to perform a viability assessment, as well as extended preservation times; in fact, due to the completely new sequence of IRI, that is, reperfusion on the device without exacerbation of the inflammatory response through the recipient’s immune cell infiltration, a novel opportunity for immunomodulation has emerged. During NMP, the donor organ regains in nearly physiologic conditions, while further damage and activation of the graft’s immune system by the recipients’ immune cells are avoided. Therefore, NMP resembles a potentially immunomodulatory phase of IRI, namely, without contact with the recipient’s immune system and corresponding risk for the hepatic graft, while the currently not fully known effect of immunomodulatory capacities of the perfusion solution and especially RBCs should be taken into account.

## 3. Hepatic IRI

IRI follows a hypoxic insult and a subsequent restoration in blood circulation, which triggers a sterile inflammatory response. During the sequence of LT, IRI and its molecular pathways represent a multifaced process. In the last decade, studies have increasingly paid attention to the role of reperfusion and the modulation of the recipient’s immune response. In particular, MP represents a sensitive phase before reperfusion that influences the dynamic of IRI within the hepatic graft.

Two distinct phases of IRI can be differentiated. First, hepatic cells are damaged during cold ischemia, which occurs when the hepatic graft is procured and subsequently stored on ice [41]. This is followed by warm ischemia, which resembles a subcategory during implantation of the hepatic graft. Here, hepatic injury is mainly due to oxygen deprivation and energy depletion [42]. The second part of IRI includes oxidative stress, inflammation, and apoptosis following reperfusion. Cell types that interact during hepatic IRI are parenchymal hepatocytes and non-parenchymal cells, such as liver sinusoidal endothelial cells (LSECs) [43], and innate immune cells such as Kupffer cells (KCs, resident liver macrophages), neutrophils, natural killer (NK) cells, natural killer T (NKT) cells, T cells, and dendritic cells (DCs) [41]. 

The underlying pathological process of cold ischemia, e.g., as part of SCS includes a change of aerobic to anaerobic metabolism with intracellular acidosis in hepatocytes. ATP depletion, intracellular ion disbalance, and the activity of proteolytic enzymes are aggravated by a cold hypoxic environment, ultimately leading to cell swelling and cell death. An increment in intracellular Ca^2+^ levels and lactic acid resembles the main aspects of hypoxic conditions, which leads to mitochondria dysfunction and reactive oxygen species (ROS) formation [44]. This is aggravated by rewarming, which results in anaerobic metabolism and further injury during reperfusion, triggering an immediate release of ROS [45]. At this stage, injury of the hepatic tissue is exacerbated, leading to a sterile inflammation due to a systemic response to inflammatory mediators. DCs, which are antigen-presenting cells (APC), complete the inflammatory response through crosstalk between innate immune cells, while LSECs show a high sensitivity to cold ischemia and trigger KC and neutrophil activation, vasoconstriction, and endothelial dysfunction [43].

Undoubtably, IRI and the underlying sterile inflammation affect the biliary tree as well. Bile duct cells show a high susceptivity for hypoxia, which is known to cause long-term complications after liver transplantation. As mentioned above, currently available data suggest a protective mechanism of D-HOPE and HOPE for developing NAS [14,15,16]. On the contrary, after NMP, no reduction of bile strictures was noted [23,39]. Interestingly, NAS seems to be a result of a multifaced process that includes the impaired regeneration of biliary epithelium by peribiliary glands (PBG) [46]. PBG represent clusters of biliary epithelial cells, interconnected with the bile duct via small canals that contain multipotent progenitor cells to support regeneration after injury, e.g., by ischemia [47]. In addition to ischemia and PGB injury, immune-driven processes play a role in the origin of NAS, since patients suffering from immune-liver disease such as auto-immune hepatitis and primary sclerosing cholangitis show an increased risk of developing NAS [46].

IRI is characterized by the activation of various molecular pathways that employ further tissue damage, such as mechanisms associated with hypoxia-inducible factor 1 alpha (HIF1 alpha) and tumor necrosis factor alpha (TNF alpha) [42]. Activation of nuclear factor kappa B (NF kappa B) was identified as a way through which TNF alpha promotes damage to hepatocytes after binding to the TNF receptor [48]. In addition to this, interleukin 1 beta (IL1b) stimulates ROS production and infiltration of neutrophils [49]. Experimental findings highlight that tissue disruption is promoted by a release of damage-associated molecular patterns (DAMPS), such as high-mobility group box 1 protein (HMGB-1), mitochondrial DNA (mt-DNA), hyaluronic acid, and heat shock proteins (HSPs) [50]. Of special interest are mechanisms within mitochondria initiating IRI by subcellular processes, including the accumulation of metabolic substrates such as mitochondrial succinate [4].

DAMPs bind to Toll-like receptors (TLRs) such as TLR4 on KCs, leading to increased activity of these resident macrophages as well as neutrophil activation within the liver [51]. This is of importance, as KCs resemble the largest group within the resident macrophage population of the human body and play a key role in the hepatic immune response. Remarkably, KCs can either enhance a proinflammatory effect or ameliorate IRI, depending on their state of polarization [52]. It is established that the degree of cellular injury (accumulated during cold and warm ischemia) predicts the severity of IRI by triggering the recipient’s immune response [53]. However, the liver itself can limit cellular injury by, e.g., clearing injured or apoptotic cells. Reperfusion without foreign immune cells on the NMP device enables KCs to phagocyte apoptotic cells and, thus, limit DAMPs, all of which presumably ameliorates the recipient’s immune response following reperfusion in the recipient. Thus, the NMP-specific subphase of reperfusion without contact with the recipient’s immune system is a completely new step in the traditional sequence of IRI.

## 4. Innate Immune Cells in the Liver

The liver contains one of the largest populations of resident innate immune cells, including NK cells, DCs, and KCs. The intrahepatic cellular compartment consists of 60–80% hepatocytes, which line the space of Dissè where hepatic stellate cells reside. Additionally, non-parenchymal cells such as LSECs line the liver sinusoids where KCs are located [54]. The unique anatomy of the liver, with a continuous antegrade arterial and portal venous blood supply, allow for the rapid influx of circulating lymphocytes, neutrophils, and monocytes as part of both innate and adaptive immune responses [55].

Immune cells can enter the liver from the systemic blood circulation via the liver sinusoids (Figure 1). Among these infiltrating cells, neutrophils represent the largest circulation fraction of leukocytes and transmigrate into the liver parenchyma once the liver is injured. Once neutrophils degranulate, they release ROS and proteases, which results in intracellular oxidative stress; mitochondrial dysfunction; and, ultimately, the cellular damage of hepatocytes [55]. Their adhesion to the epithelium is, among others, mediated by interactions with intercellular adhesion molecules (ICAMs) and is known to be associated with acute liver rejection [56]. In contrast, neutrophil-elicited phagocytosis is essential for tissue repair and protection, which further demonstrates their role as being context dependent.

Distributed mainly in the perivenular and portal space, DCs are leukocytes that link innate and adaptive immune responses [57]. DCs interact with T, NK, and NKT cells, while the latter might be correlated with IL10 and TGF-beta expression in an immunoregulatory way [58]. Hepatic NK and NKT cells represent the predominant lymphocyte population. They communicate with both the innate and the adaptive immune systems and account for more than 50% of hepatic lymphocytes [59]. Located next to KCs in the liver sinusoids, NK and NKT cells interact by cytokines such as IL12 and IL18, as well as interferon gamma (IFN gamma). Interestingly, these resident, hepatic NK cells obtain immunomodulatory capacities and act in favor of immune tolerance in mice [60]. On the contrary, facing IRI and cellular damage, circulatory NK cells promote injury of hepatocytes and cell death mainly by IFN gamma [59]. During LT, recipient-derived NK cells have been shown to lead to graft rejection, with increased graft survival after circulatory NK cell depletion, while donor-derived NK cells play a role in tolerance induction [61]. Of note is that IFN-gamma is obligatory for the development of resident NK cell populations, highlighting its context-dependent role [60].

The resident cell population of KCs, which were recently believed to be exclusively sessile, can either be resident or monocyte derived, inheriting a broad panel of functions within the liver. They reside with LSECs, DCs, and NKs along liver sinusoids and interact with T cells, hepatocytes, stellate cells, and neutrophils as well. In addition to phagocytosis and presenting antigens, KCs release a variety of cytokines, such as TNF-alpha, IL6, IL1, IL10, IL13, and ROS [62]. As outlined earlier, innate immune cells are part of complex cellular interactions concerning hepatic homeostasis and the regulation of the immune response. Among these innate immune cells, KCs undoubtedly are key players in the modulation of inflammation and the crosstalk between components of the immune response.

## 5. Immune-Modulating Mechanism of MP

Jassem et al. [33] used gene expression analysis in pre- and post-reperfusion biopsies to demonstrate NMP-elicited IRI mitigation. Mechanisms behind IRI mitigation during NMP were mainly identified as the downregulation of pro-inflammatory cytokines, which leads to fewer CD4- and CD8-positive effector T cells, while anti-inflammatory regulatory T-cell (Treg) subpopulations maintained or even increased their number [33]. Interestingly, the group highlighted that INF gamma affects T cells, while it is known to classically stimulate M1 macrophage polarization. Additionally, the group found decreased neutrophils in hepatic grafts after NMP, which contributes to anti-inflammatory conditions. Other studies have concordantly shown decreased pro-inflammatory cytokines such as TNF-alpha, IL6, and IL1 after NMP in an animal model [63]. TNF-alpha has long been known as the central mediator of hepatic injury. Among others, it is released by KCs, which have been identified as being a mass producer of TNF-alpha. A decrease in TNF-alpha ameliorates oxidative stress, KCs, and neutrophil activation. ROS itself is a major stimulus for the release of HMGB-1, which binds to TLR4 on KCs and DCs, leading to the production of IL6, TNF-alpha, and ICAM1 [64].

The mitigation of ROS and KC activation was furthermore detected during HOPE, wherein a decrease in HMGB-1 release was demonstrated in porcine livers [10]. Further experimental animal studies suggest that measuring inflammatory cytokines and DAMPs during MP could lead to an adequate viability assessment [65]. The authors put emphasis on the NLR family pyrin-domain-containing 3 (NLRP3) inflammasome activation as a parameter for liver cell apoptosis. During NMP, heme-oxygenase-1-modified bone marrow mesenchymal stem cells were applied, leading to a decreased expression of the HMGB-1 and TLR4 pathways, suggesting a decreased pro-inflammatory macrophage response [66].

### Effects of MP on Macrophages

As outlined above, both hypothermic and normothermic MP led to an ameliorated IRI as well as a decreased pro-inflammatory cytokine, and KC activation was described. Of remark is that studies directly characterizing KCs are rarely available; rather, markers that identify macrophage activation are analyzed (Table 1). One of the studies that directly identified macrophages is an early rodent experiment on MP that described a slightly decreased colloidal carbon uptake within continuous perfused liver grafts (bubble oxygenated University of Wisconsin (UW) solution) [67]. Although a clinical study on HMP histologically identified fewer macrophages in HMP biopsies compared to SCS, a quantification was not conducted [68]. In addition to that, the cytokines TNF-alpha, IL1, and IL8 were reduced in the HMP group. One aspect of IRI mitigation in MP might be an overall reduction of macrophages as a result of washing out [69]. Nevertheless, a following small animal study detected an activation of cells that expressed KC surface markers, while a differentiation between pro- and anti-inflammatory subtypes was not conducted [70]. Since NMP can be used as delivery tool, within this study, promising results regarding the suppression of KC activation by the application of TGB-beta and IL-10 during perfusion were obtained. As indirect parameters for KC activation for both hypothermic and normothermic MP, a reduction of cytokines was reported compared to SCS [10,33,71]. The mitigation of IRI and reduction of pro-inflammatory cytokines that are related with KC activation was detected in an porcine HOPE model [10], a porcine split liver NMP model [71], and a clinical trial for NMP [33].

## 6. Innate Immune Cells in Hepatic IRI

### 6.1. Neutrophils in Hepatic IRI

As pointed out earlier, the actions of neutrophils are far from being one sided, and their capacities do not work only in favor of cell damage and inflammation. Their phagocytotic functions are essential for clearing cell debris and performing tissue repair [72]. Despite the clear evidence that massively recruited neutrophils during hepatic IRI lead to the death of hepatocytes and ROS production, a subpopulation of neutrophils might benefit an earlier regeneration. Their beneficial role is partly due to the promotion of a macrophage phenotype switch from pro-inflammatory macrophages (Ly6C^high^ CX3CR1^low^) to reparative macrophages (Ly6C^low^, CX3CR1^high^), as suggested in acute liver injury [73]. As recently investigated, neutrophil-derived microRNA-233 plays a crucial role in driving macrophage polarization to a less inflammatory phenotype, which is, among others, associated with reduced NLRP3 inflammasome expression [74]. In both microRNA-233- and neutrophil-depleted mice, the presence of pro-inflammatory macrophages, hepatic inflammation, and fibrosis was detected [74]. Additionally, neutrophils interacting with NK cells lead to the increased secretion of IFN gamma by NK cells and prolonged survival of neutrophils [75]. In summary, in hepatic IRI, neutrophils may engage in various multidirectional communications with other innate immune cells such as NK cells and KCs.

### 6.2. NK and NKT Cells in Hepatic IRI

NK and NKT cells originate from the bone marrow and then differentiate in the spleen or thymus, after which they partly reside in the liver or remain circulatory. While NK cells are mass producers of IFN-gamma in liver diseases such as hepatic IRI, NKT cells either secrete IFN-gamma or IL4, depending on their present phenotype [76]. As described earlier, IFN-gamma plays a role in hepatic IRI, leading to further inflammation and tissue damage, as well as early graft rejection [77]. In contrast to other lymphocyte populations such as B and T cells, NK and NKT cell activation is not due to the stimulation of one or two main receptors but is regulated by an ensemble of activating and inhibiting signals [78]. In hepatic IRI, receptors that are primarily engaged in NK cell activation are known to be NK granule 7; CD16; CD226; NKG2D; and NKp46, -44, and -30, while the killer cell immunoglobulin receptor (KIR), Ly49 receptor in mice, CD96, T-cell immunoreceptor with Ig and ITIM domains (TIGIT), and lymphocyte activation gene 3 (LAG3) are described as being inhibitory receptors [58].

Experimental data revealed that during the initial phase of IRI in cold ischemia, NK cells significantly increased in the liver parenchyma but did not participate in tissue damage. During the following warm IRI, NK cell activation by NKG2D, along with increased inflammatory cytokines such as TNF-alpha, IL1, and IL6, was observed [58]. On the other hand, there was an upregulation of the TRAIL ligand in NK cells after IRI, which led to decreased neutrophil infiltration, IL6, and transaminase level, suggesting that NK cells play a role in immune tolerance after transplantation [79].

Additionally, NKT cells might have distinct roles regarding hepatic IRI as well. A blockade of NKT cell activation by CD1d, which is expressed by hepatocytes, resulted in a reduction in the IRI, as IFN-gamma-mediated neutrophil accumulation and tissue damage were reduced [80]. Moreover, an amelioration of hepatic IRI was achieved by activating type II NKT cells subsequent to sulfatide activation, thus uncovering their immunoregulatory capacities [81]. Together, both NK and NKT cells aggravate the innate immune response to hepatic IRI through interactions and the stimulation of other innate immune cells such as KCs, neutrophils, or their own cell population, while a reduction in T cells has no effect on IRI [82]. As pointed out, these interactions are accessible by various factors and can be modulated in an immunoregulatory way.

### 6.3. Macrophage Polarization in Hepatic IRI

Several studies have shown that during IRI, macrophages are activated and promote cellular damage by ROS production and the secretion of pro-inflammatory cytokines [49,83,84]. However, this response of macrophages in the sequence of IRI is only a single-sided view, as macrophages can change between many sub-polarizations, which either act in favor of inflammation or perform immunomodulatory, anti-inflammatory functions [64,83,85]. Among other innate immune cells, the liver contains both resident macrophages such as KCs and non-resident blood-monocyte-derived macrophages (MoMFs). KCs are located along the hepatic LSECs and inherit a mandatory role in the clearance of pathogens and iron metabolism within the liver, which maintains local homeostasis. Interestingly, the self-renewing capacities of KCs, which are derived from fetal-liver-derived red bone marrow progenitor cells and yolk-sac-derived red bone marrow progenitor cells, are impaired by ischemia, leading to a depletion of KCs [86]. After KC depletion, infiltrating MoMFs show the capacity to differentiate into KCs [87]. In this scenario, however, ROS and cytokine production from apoptotic KCs were described to promote pro-inflammatory MoMF polarization [86]. The recent progress in macrophage characterization utilizing single-cell RNA sequencing revealed a dynamic balance of several macrophage polarization states [88]. Therefore, the known dogma of M1- and M2-like macrophages needs to be reconsidered [86]. The described pro- and anti-inflammatory subpopulations matched either the recently identified M1 or M2 phenotype or both. M1-like macrophages were formerly known to display pro-inflammatory features, such as the secretion of TNF, IL-1 beta, or ROS and the expression of inducible nitric oxide synthase (iNOS) [85]. M2-like macrophages act rather immunomodulatory by the secretion of IL-10 or due to their high phagocytic capacity and the expression of arginase 1 [89]. Nevertheless, several subtypes of macrophages have been identified as showing either M1- or M2-like polarization or both. Therefore, it appears far more practicable to describe macrophages by their origin and activating stimuli. KCs are known to express a CD45+ F4/80+ CD11b^int^ phenotype, and they can be differentiated from MoMFs, which show a Ly6C-positive subset in experimental studies. Conversely, Ly6C^high^ MoMFs are bone marrow derived and usually express inflammatory chemokine receptors such as the chemokine (C-C-motif) receptor (CCR) 2 [90]. The spleen is a source of Ly6C^low^ MoMFs, showing phagocytic and patrolling capacities [91] and failing to infiltrate the liver. Moreover, peritoneal-cavity-derived macrophages expressing GATA6 were shown to infiltrate the injured liver in mice. GATA6-positive macrophages facilitated tissue repair and performed phagocytosis [92]. While equal resident macrophages in the peritoneal cavity have been characterized in humans [93], further studies regarding liver infiltration and tissue repair in humans are required. KCs attract MoMFs and NK cells by massive chemokine (C-C motif) ligand (CCL) 2 expression. During IRI, KCs not only interact within their own cell population but also with neutrophils and T cells through chemokine secretion. Interestingly, KCs play a role in neutrophil and T cell polarization and can promote immunoregulatory neutrophil and T cell subtypes. Regarding IRI, pro-inflammatory stimuli lead to an activation of the inflammasome due to an upregulation of pro-IL1 beta and pro-IL18 within KCs, which results in the secretion of IL1 beta and IL18 [83]. Inflammasome formation within KCs, which was recently identified as NOD-, LRR-, and pyrin domain-containing 3 (NLRP3) inflammasome, is promoted by the recognition of DAMPs [90]. These are derived from injured hepatocytes and include ATP, cholesterol crystals, DNA fragments, uric acid, and fatty acids. Inflammasome formation was also identified in IRI induced by ROS and HMGB1 via TLR4, while NLRP3 inflammasome activation via ATP again leads to the release of IL1B and HMGB1 [90].

### 6.4. Macrophages and Their Role in the Resolution of Inflammation

In their pro-inflammatory state, KCs promote the activation of complement mediators such as C3a and C5a. CD14+, CLEC5A+, and S100A9+ were identified as gene subsets of pro-inflammatory macrophages in the human liver [94]. Cytokines such as TNF-alpha are secreted by KCs, which subsequently enhance the expression of intracellular adhesion molecule 1 (ICAM1) and vascular adhesion molecule 1 (VCAM1), allowing further immune cell migration, especially of neutrophils [44]. On the contrary, macrophages that inhibit further inflammation and act in favor of liver restoration show CD163+, MERTK, and CD16+ gene subsets [95]. IL10 was recognized as an anti-inflammatory cytokine secreted by KCs, which suppress NF-kappa-B-associated actions and inhibit pro-inflammatory macrophage polarization. Moreover, IL10 was identified as promoting anti-inflammatory polarization, which counteracts pro-inflammatory KC polarizations [52]. Additionally, IL10 was found to inhibit NK-cell-promoted inflammatory stimuli [90]. Other soluble factors such as IL4 and TGF-beta lead to a change in KC polarization toward immunomodulatory states as well [83]. As stated earlier, the switch in KC subtypes resembles a dynamic balance. One way by which immunomodulatory subtypes lead to a resolution of inflammation is the phagocytosis of damaged hepatocytes, KCs, and LSECs, which reduces pro-inflammatory stimuli, such as DAMPs (Figure 2). The differentiation process of KCs and infiltrating macrophages toward a phenotype that promotes the resolution of inflammation appears to be augmented by macrophage-colony-stimulating factor 1 (CSF1). For monocyte-derived macrophages, this process was shown to include a switch from CCR2+ and CX3CR1- toward CCR2- and CX3CR1+ gene subsets. Under anti-inflammatory stimuli, monocyte-derived macrophages produced vascular endothelial growth factor-alpha (VEGFA) and, thus, promoted the restoration of the vasculature. The pivotal role of macrophages in the resuscitation of liver injury was shown by the prolonged repair phase in mice that lacked infiltrating monocytes. Furthermore, studies on animals have indicated that KC subtypes show the capacity to inhibit pro-inflammatory CD4 T cells through high secretion of IL10 and TGF-beta [96]. Moreover, stimulation of anti-inflammatory Tregs was shown. These findings are consistent with other animal disease models that have demonstrated the anti-inflammatory immune regulation of T cells by regulatory macrophages. Nevertheless, the meaning of KC and T-cell interaction for IRI in liver transplantation needs to be further elucidated. Similar to macrophage plasticity, several studies indicate that neutrophils can switch between a pro- and anti-inflammatory subtype. During inflammation, KCs guide neutrophils by chemokine (C-X-C motif) ligand (CXCL) 1 and CXCL2 to the luminal surface of LSECs [97]. The specific microenvironment, including activation and communication with neutrophils, drives macrophage polarization in one direction or the other. For tissue repair and regeneration in sterile inflammation, neutrophils are essential, as they perform phagocytosis and remove damaged tissue. Moreover, secretion of Arginase 1 by neutrophils shows potential for inhibiting T cell cytotoxicity. Next, phagocytosis of apoptotic neutrophils by macrophages leads to the resolution phase of inflammation [97].

Macrophage polarization is thus shifted to an anti-inflammatory state, in which an upregulation of IL1R, Il10, and TGF-beta secretion occurs, leading to tissue repair and immune tolerance initiated by Tregs [98]. T cells complementarily interact with KC and lead towards anti-inflammatory activation states by IL4, TIM-3, and its galectin-9 (Gal-9) ligand [99].

## 7. Conclusions and Future Perspectives

As IRI-associated events such as graft dysfunction, organ failure, and rejection still challenge LT to date, the resolution of reperfusion injury remains a relevant topic in clinical practice. The innate immune system plays a major role in ameliorating IRI. KCs, as central components of sterile inflammation, are influential in the immunoregulation of IRI due to their multifaceted capacities. Pro-inflammatory and anti-inflammatory polarization states of KCs appear to be a dynamic equilibrium that should be further guided toward anti-inflammatory directions.

During MP, a unique time slot is established, wherein phagocytotic and immunoregulatory directions of macrophage polarization can be employed. Recent studies have already identified mechanisms of action regarding IL10-guided promotion of anti-inflammatory CD163+, MERTK, and CD16+ gene subsets. Moreover, the initiation of a resolution of inflammation was recently demonstrated to be led by neutrophil–macrophage crosstalk. The communication between regulatory macrophages and Tregs also appears to be a factor in the resolution of tissue damage. MP offers the opportunity to modulate these immune responses toward resolution due to lymphocyte-free perfusion, oxygen, and energy reinforcement. Apart from decreasing pro-inflammatory stimulators such as HMGB1, TNF-alpha, and IL6 during MP, further anti-inflammatory preconditioning could be established by the application of anti-inflammatory mediators. Consequently, novel mediators such as the recently discovered subtypes of peritoneal-cavity-derived macrophages could be applied during MP and may lead to the reconditioning of immune cell polarization. Additionally, progress has already been made by using heme-oxygenase-1-modified bone marrow mesenchymal stem cells to reduce HMGB1 expression and TLR4 pathway expression, which suggests reduced inflammatory stimuli. Therefore, therapeutic strategies targeting innate immune cells, particularly macrophages and neutrophils, warrant broad clinical and experimental attention to overcome IRI in liver transplantation.

## Figures and Tables

**Figure 1 jcm-11-06669-f001:**
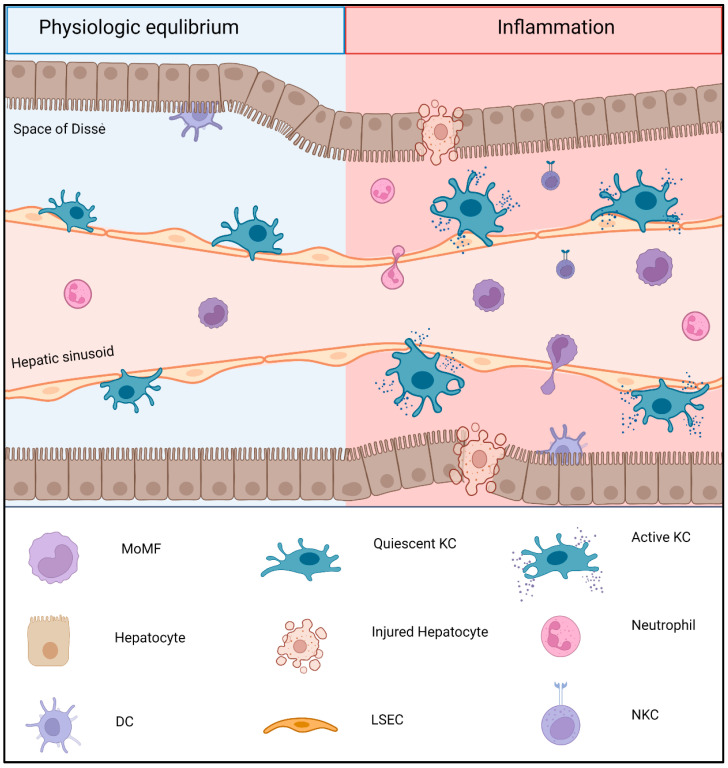
Schematic overview of Kupffer cell (KC) activation states. During the physiologic equilibrium, KCs remain quiescent and self-maintaining, located next to liver sinusoidal endothelial cells in the space of Dissè, which separates hepatocytes from sinusoidal endothelium. In the absence of inflammatory stimuli, only a small number of monocyte-derived macrophages (MoMFs) are recruited from the hepatic sinusoids. Following an inflammatory stimulus (e.g., ischemia reperfusion injury), a cascade of proinflammatory stimuli and damaged hepatocytes lead to an increasing macrophage activation, large numbers of MoMFs, and neutrophils invading the hepatic tissue. The interaction of dendritic cells, natural killer cells, neutrophils, and KCs aggravate inflammation and tissue damage. (Created with BioRender.com, accessed on 24 September 2022).

**Figure 2 jcm-11-06669-f002:**
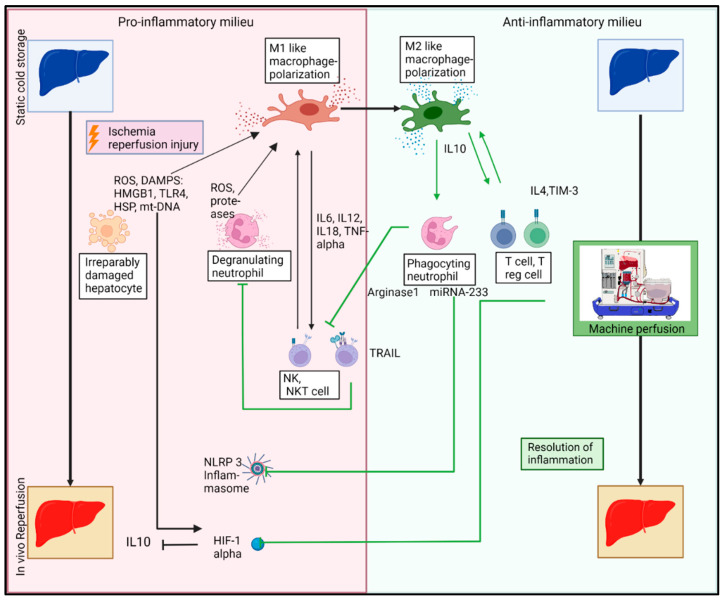
Interaction of innate immune cells during machine perfusion (MP) mitigates hepatic ischemia reperfusion injury (IRI). During MP, a novel phase of innate immunity is to be established. Macrophages and their subsets can be considered as game changers that drive innate immunity towards an anti-inflammatory milieu. Interleukin 10 (IL10) secreted by M2-like macrophages inhibit the dynamic change toward M1-like macrophage polarizations. The establishment of an anti-inflammatory milieu affects associated innate immune cells such as neutrophils that clear cell debris of irreversibly damaged hepatocytes together with macrophages. Innate immune cells initially known to act in a primarily pro-inflammatory way were found to promote anti-inflammatory effects as well. MicroRNA-233 secretion by neutrophils inhibits NLR-family-pyrin-domain-containing 3 (NLRP3) inflammasome formation. Natural killer (NK) cells and natural killer T (NKT) cells inhibit counteract neutrophils via the tumor-necrosis-factor-related apoptosis-inducing ligand. Moreover, MP mediated the amelioration of hypoxia-inducible factor 1 (HIF1)-alpha, reducing proinflammatory pathways. The interlink between M2-like macrophage polarizations and regulatory T (T reg) cells through a secretion of transforming growth factor-beta and IL10 supports macrophage-mediated anti-inflammatory capacities and T-cell-mediated immune tolerance. (Created with BioRender.com, accessed on 24 September 2022).

**Table 1 jcm-11-06669-t001:** Effects of MP on KC.

Study	Type of MP	Analysis Criteria	Results
Southard et al. [67] Rodent animal model	Oxygenated UW solution	Colloidal carbon uptake	Reduced KC activation
Henry et al. [68] Clinical trial	HMP, 4.2 h mean perfusion time	Biopsy, qPCR	Fewer KCs in fluorescence staining Reduced pro-KC activating cytokines
Vries et al. [69] Rodent animal model	SNMP, 3 h perfusion time	Imaging flow cytometry	Percentage of KCs (CD14+/CD105 surface markers)
Carlson et al. [70] Rodent animal model	NMP, 4 h perfusion	Flow cytometry, ELISA	Percentage of KCs (CD40, CD86, and PD-L1 surface markers), cytokines
Schlegel et al. [10] Porcine animal model	HOPE, 1 h perfusion	Immunofluorescence, ELISA	Cytokines
Zhang et al. [71] Porcine animal model	NMP, split liver (NA perfusion time)	ELISA, Western blotting	Cytokines
Jassem et al. [33] Human clinical trial	NMP, 20 liver grafts (NA perfusion time)	Flow cytometry, immunohistochemistry	Cytokines

## Data Availability

Not applicable.

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
