# Peer review of "Innate Immune Cells during Machine Perfusion of Liver Grafts—The Janus Face of Hepatic Macrophages"

_jcm, 2022, doi:10.3390/jcm11226669_

Round 1

Reviewer 1 Report

Dear authors, the article presented by you is devoted to a topical and important problem of our time. The article is very informative and full of modern vision of the problem and facts of experimental and clinical studies. I would also like to note the structuredness and interesting presentation of the material by the authors.

Several questions came up while reading.

I would like to clarify whether the expression in line 23 “…the preferred treatment of hepatic malignancies” is understood correctly, because in case of malignant tumor processes, organ transplantation, in particular, the liver, is not performed. Perhaps you meant benign tumor processes.

I would like to recommend the authors to make a slight clarification regarding the relationship between the processes of ischemia-reperfusion liver injury and the development of non-anastomotic biliary strictures, which lead to fibrous narrowing of the lumen of the bile ducts and obstruction of the outflow of bile during organ transplantation. Please tell me, is the role of macrophages and other cells of innate immunity in the pathogenesis of non-anastomotic biliary tract strictures known? Because evidence from a multicentre controlled study suggests a lower risk of non-anastomotic biliary strictures after liver transplantation obtained after circulatory death when using hypothermic oxygenated perfusion compared with using static cold preservation (doi: 10.1056/NEJMoa2031532). I wonder if there have been studies aimed at identifying the connection between the activation of immune system cells and organ transplantation, with different preoperative preparations?

In my opinion, Chapter 4. Innate Immune Cells in the Liver can (should) emphasize the origin and specific phenotype of resident hepatic NK cells (as opposed to those circulating in the peripheral blood) influencing their functional potential (doi:10.1016/bs. ai.2019.11.002). Lines 192-193 note the main damaging effect mediated by NK cells when IRI develops against the background of the action of IFN-y. At the same time, it was noted that IFN-y is a necessary factor in the development of resident liver populations of NK cells and the formation of their protective properties (in mice) (doi: 10.1016/bs.ai.2019.11.002).  

The review discusses the predominantly positive aspects of machine perfusion of liver grafts, which can be reduced to a change in the polarization of macrophages into an anti-inflammatory phenotype. Potential difficulties and risks are not considered here. Thus, the issue of selection of perfusion fluid and oxygen carrier, their interaction with immunocompetent cells, and in particular with macrophages, is completely omitted. The erythrocytes of another donor are most often used as an oxygen carrier, which is also an additional risk factor, which is a default figure in this article.

  The data described suffer from some generalizations. So it is not entirely clear what is the minimum required perfusion time for the implementation of the processes described in the review. A similar question is regarding the maximum exposure time after which negative effects (for example, due to the destruction of erythrocytes and an increase in the load on macrophages, which must also eliminate them) begin to exceed positive ones.

Author Response

Dear Reviewer 1,

We thank you very much for your detailed and extraordinary remarks and provided a point-by-point discussion within our response letter. We welcome this opportunity to develop and revise our manuscript.

Please see the attached letter (as per below) 

Reviewer 2 Report

This review  describes the mechanism of the role of innate immunity in hepatic ischemia-reperfusion injury and demonstrates the effect of mechanical perfusion on innate immunity. However, not much is said about the effect of mechanical perfusion on macrophages. Therefore, it is suggested to delete "The Janus face of hepatic macrophages" from the title to avoid misleading.

Author Response

Dear Reviewer,

Thank you very much for your excellent remarks. We provided a detailed response letter (please s. as per below).

Best Regards
